# The SQSTM1-NUP214 fusion protein interacts with Crm1, activates *Hoxa* and *Meis1* genes, and drives leukemogenesis in mice

**Catherine P. Lavau**[1]\*, **Waitman K. Aumann**[2,3], **Sei-Gyung K. Sze**[4], **Veerain Gupta**[5], **Katelyn Ripple**[5], **Sarah A. Port**[6], **Ralph H. Kehlenbach**[7], **Daniel S. Wechsler**[2,3]\*

**1** Department of Neurosurgery, Duke University Medical Center, Durham, North Carolina, United States of America, **2** Aflac Cancer & Blood Disorders Center, Children's Healthcare of Atlanta, Atlanta, Georgia, United States of America, **3** Department of Pediatrics, Emory University School of Medicine, Atlanta, Georgia, United States of America, **4** Maine Children's Cancer Program, Scarborough, Maine, United States of America, **5** Department of Pediatrics, Duke University Medical Center, Durham, North Carolina, United States of America, **6** Department of Molecular Biology, Princeton University, Princeton, New Jersey, United States of America, **7** Department of Molecular Biology, Faculty of Medicine and the Göttingen Center for Molecular Biosciences (GZMB), Göttingen, Germany

\* catherine.lavau@duke.edu (CPL); dan.wechsler@emory.edu (DSW)

**Data Availability Statement:** All relevant data are within the manuscript and its Supporting Information files.

## Abstract

The NUP98 and NUP214 nucleoporins (NUPs) are recurrently fused to heterologous proteins in leukemia. The resulting chimeric oncoproteins retain the phenylalanine-glycine (FG) repeat motifs of the NUP moiety that mediate interaction with the nuclear export receptor Crm1. NUP fusion leukemias are characterized by *HOXA* gene upregulation; however, their molecular pathogenesis remains poorly understood. To investigate the role of Crm1 in mediating the leukemogenic properties of NUP chimeric proteins, we took advantage of the Sequestosome-1 (SQSTM1)-NUP214 fusion. SQSTM1-NUP214 retains only a short C-terminal portion of NUP214 which contains FG motifs that mediate interaction with Crm1. We introduced point mutations targeting these FG motifs and found that the ability of the resulting SQSTM1-NUP214$^{FGmut}$ protein to interact with Crm1 was reduced by more than 50% compared with SQSTM1-NUP214. Mutation of FG motifs affected transforming potential: while SQSTM1-NUP214 impaired myeloid maturation and conferred robust colony formation to transduced hematopoietic progenitors in a serial replating assay, the effect of SQSTM1-NUP214$^{FGmut}$ was considerably diminished. Moreover, SQSTM1-NUP214 caused myeloid leukemia in all transplanted mice, whereas none of the SQSTM1-NUP214$^{FGmut}$ reconstituted mice developed leukemia. These oncogenic effects coincided with the ability of SQSTM1-NUP214 and SQSTM1-NUP214$^{FGmut}$ to upregulate the expression of *Hoxa* and *Meis1* genes in hematopoietic progenitors. Indeed, chromatin immunoprecipitation assays demonstrated that impaired SQSTM1-NUP214 interaction with Crm1 correlated with impaired binding of the fusion protein to *Hoxa* and *Meis1* genes. These findings highlight the importance of Crm1 in mediating the leukemogenic properties of SQSTM1-NUP214, and suggest a conserved role of Crm1 in recruiting oncoproteins to their effector genes.

**Funding:** WKA - ASH Research Training Award for Fellows; Hyundai Hope On Wheels Young Investigator Grant. SKS - NHLBI T32 5T32HL007057-40. RHK - German Research Foundation (RHK; SFB860). DSW - Hyundai Hope On Wheels Scholar Award, St. Baldrick's Foundation Research Award; Schiffman Family Foundation. CPL - Hyundai Hope on Wheels Grant; INSERM scientist. The funders had no role in study design, data collection and analysis, decision to publish, or preparation of the manuscript.

**Competing interests:** The authors have declared that no competing interests exist.

## Introduction

Nucleoporins (NUPs) are components of nuclear pores–multiprotein channels in the nuclear envelope that control the transfer of macromolecules between the nucleus and cytoplasm. There are approximately 30 different NUPs, some of which form the scaffold of the nuclear pore; other NUPs are positioned in the channel of the pore and regulate the traffic of macro-molecules [1]. These latter NUPs, referred to as FG-NUPs, contain multiple phenylalanine and glycine (FG)-repeat units that form short clusters of hydrophobic residues separating long stretches of hydrophilic amino acids. The FG repeats provide docking sites for transport receptors as they move cargo across the nuclear pores; notably, they mediate interaction with the nuclear export receptor CRM1 (chromosome region maintenance 1, also known as exportin 1 or XPO1). Two FG-NUPs, NUP98 and NUP214, have been identified in recurrent chromosomal translocations that result in their fusion to heterologous proteins [2, 3]. These chimeric proteins have leukemogenic properties, causing predominantly acute myeloid leukemia (AML) or T-cell acute lymphoid leukemia (T-ALL) that generally have a poor prognosis.

A large number of chromosomal translocations affecting *NUP98* have been described, and more than 30 different partner proteins have been reported (reviewed in [4, 5]). A constant feature of the various NUP98 fusion proteins is the conservation of FG repeats contained in the amino terminal region of NUP98, which is fused in frame to the carboxy terminus of the heterologous proteins. In the case of NUP214, four different fusion proteins have been reported in patients with leukemia: SET-NUP214, DEK-NUP214, SQSTM1-NUP214, and NUP214-ABL1. Each of these fusions conserves some or most of the FG repeats present in the carboxy half of NUP214 [6, 7]. A common feature of leukemias associated with these NUP98 and NUP214 fusions is the overexpression of *HomeoboxA* (*HOXA*) genes [3, 7]. *HOX* genes encode for homeobox transcription factors that specify cell identity in early development and subsequently regulate multiple processes, including hematopoiesis. *HOXA* genes in particular are critical effectors of leukemogenesis and their expression is upregulated by various oncoproteins such as MLL fusion proteins, mutated NPM1 and CALM-AF10 [8, 9].

Several non-mutually exclusive mechanisms have been proposed to explain the leukemogenic properties of NUP fusion proteins. In contrast to native NUPs which are predominantly localized at the nuclear rim, a number of NUP fusion oncoproteins are concentrated in nuclear subdomains in which CRM1 is also present. This mislocalization of CRM1 is believed to impair its ability to export cargo proteins from the nucleus to the cytoplasm [10–12]. Hence, NUP fusion proteins have been proposed to contribute to oncogenic transformation by perturbing the cellular distribution of specific transcription factors, whose deregulation could contribute to leukemogenesis [12].

The oncogenic activity of NUP fusions has also been proposed to result from hijacking the ability of NUPs to regulate transcription. In addition to its role at nuclear pores, a portion of NUP98 localizes to the nucleoplasm and binds promoters to regulate the transcription of adjacent genes [13, 14]. In particular, NUP98 was found to control the expression of developmental genes in mammalian cells [13] and of *HOX* gene homologs in Drosophila [15]. This transcriptional role of NUP98 is consistent with the ability of the FG repeat containing domain of NUP98 to interact with transcriptional co-activators such as CREB-binding protein (CBP)/p300 [16] or repressors such as HDAC1 [17, 18]. Fusion with a heterologous protein containing a DNA binding domain, such as the homeobox domain found in several NUP98 partner proteins, generates a chimeric transcription factor that can induce the overexpression of *HOXA* cluster genes [19]. Further indication that NUP fusion proteins can directly activate the expression of *HOXA* was provided by chromatin immunoprecipitation (ChIP) assays demonstrating that several NUP98 fusion proteins, as well as SET-NUP214, bind to *HOXA* gene loci [3, 18,

20–23]. In the case of NUP98-NSD1, NUP98-PHF23 and NUP98-JARID1A, the interaction with *HOXA* genes was shown to be mediated by the chromatin binding domain of the fusion partner proteins. More recently, the binding of NUP98 fusion proteins to *HOXA* genes was shown to be mediated by the NUP98 moiety through its interaction with proteins of the NSL/MLL1 complexes; this suggests that NUP98 and MLL fusion proteins might use a common pathogenic mechanism to induce leukemias [23]. In contrast to NUP98 fusions, the mechanisms underlying the recruitment of NUP214 fusion proteins to *HOXA* genes have yet to be elucidated.

We have previously shown that the ability of CALM-AF10 to activate *Hoxa* genes is mediated by Crm1 [24]. CALM-AF10 interacts with Crm1 via a Nuclear Export Signal (NES) contained in the CALM moiety, and mutation of the NES or pharmacological inhibition of the Crm1/NES interaction abolishes the ability of CALM-AF10 to bind to and activate the transcription of *Hoxa* genes. We found that Crm1 occupies *Hoxa* chromatin and proposed that it can tether CALM-AF10 to the loci of its effector genes. Subsequent Oka *et al*. showed that Crm1 similarly enables the recruitment of the NUP98-HOXA9 fusion protein to the regulatory region of *Hox* genes [20]. Very recently these authors also showed that SET-NUP214 and mutated NPM1 bind their *HOX* target genes via CRM1 [25].

In the present study, we set out to further study the involvement of Crm1 in the oncogenicity of NUP fusion proteins by focusing on the SQSTM1-NUP214 fusion protein. The *SQSTM1-NUP214* translocation has only been reported in two patients, one with AML and one with T-ALL [26, 27]. The *SQSTM1* gene encodes the adaptor protein p62 –a ubiquitin binding protein that targets proteins for degradation by autophagy–and SQSTM1-NUP214 contains the amino-terminal portion of p62 fused to a small segment of NUP214. The advantages of studying SQSTM1-NUP214 are its small size (374 amino acids [27]) which facilitates biochemical studies, and the fact that it retains a short (121 amino acids) carboxy terminal region of NUP214 that encompasses only 14 of the 44 FG motifs contained in NUP214. This region overlaps with the NUP214 fragment that was used to identify FG motifs interacting with CRM1 by crystallography (illustrated in Fig 1A) [28]. Five of these CRM1-interacting FG motifs are present in the SQSTM1-NUP214 fusion protein. To investigate the role of Crm1 in leukemogenesis, we mutated those FG motifs and found a 50% reduction in the ability of the resulting SQSTM1-NUP214$^{FGmut}$ fusion protein to bind Crm1 in transduced murine embryonic fibroblasts (MEFs). Impaired Crm1 binding assayed in MEFs correlated with loss of leukemogenicity in transplanted mice, reduced ability of SQSTM1-NUP214 to activate *Hoxa* and *Murine ecotropic viral integration site 1* (*Meis1*) genes in hematopoietic primary cells and reduced binding to *Hoxa* and *Meis1* genes in MEFs. These findings suggest a critical role of Crm1 in *SQSTM1-NUP214* leukemogenesis.

## Materials and methods

### Plasmids, cell lines and cell culture

Overlapping fusion PCR was used to create the *SQSTM1-NUP214$^{FGmut}$* construct, using *SQSTM1-NUP214* and *NUP214$_{1859-2090}$X3* sequences as PCR templates [10, 28]. SQSTM1-NUP214$^{FGmut}$ contains the F1982S, F1988S, F2012S, F2024S, and F2026S mutations. PCR was used to fuse an HA tag at the amino-terminal end of SQSTM1-NUP214 and SQSTM1-NUP214$^{FGmut}$, and to add compatible restriction sites to clone the fusion genes into the MSCV-IRES-eGFP and MSCVpuro backbones.

Retroviral particles were generated using Plat-E cells [29] as previously described [24]. Murine embryonic fibroblasts (MEFs) obtained from the ATCC (CRL-2991) were grown in

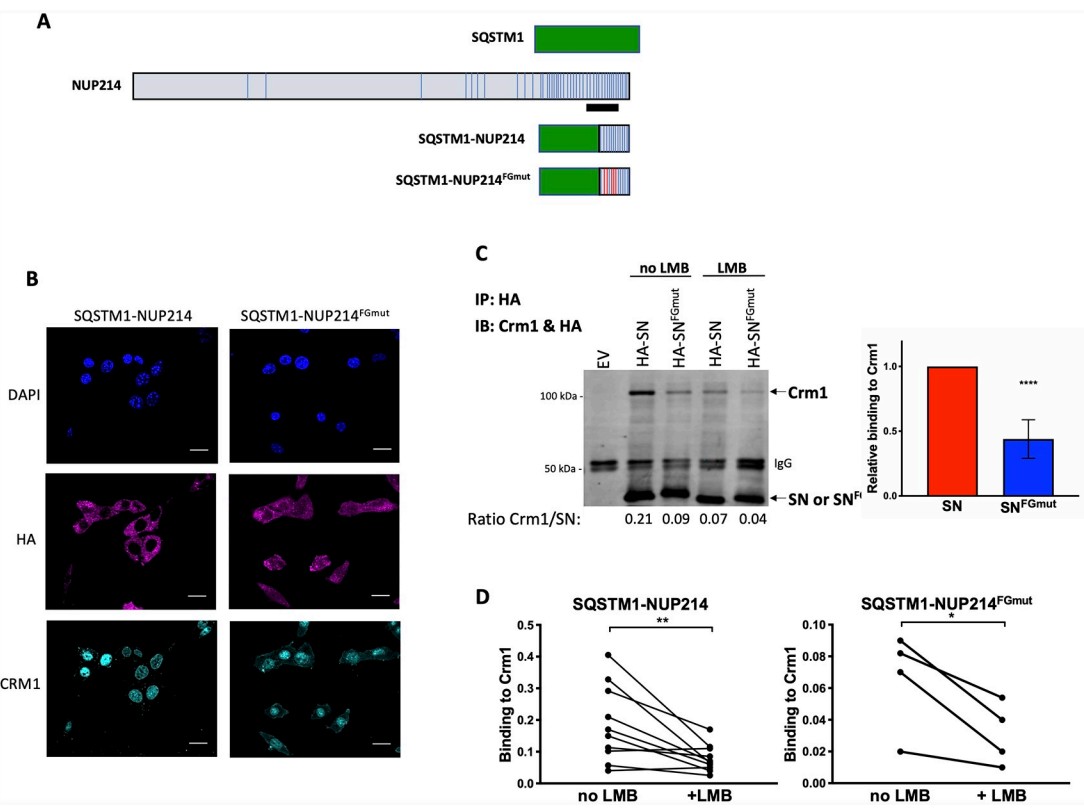

**Fig 1. Mutation of NUP214 FG repeats impairs the interaction of SQSTM1-NUP214 with Crm1.** (**A**) Schematic representation of SQSTM1, NUP214, SQSTM1-NUP214 and SQSTM1-NUP214^FGmut proteins. FG repeats are shown as vertical bars. The black line denotes the NUP214 fragment (amino acids 1916–2033) used in the crystal structure analysis of the Crm1/NUP214 complex which identified the FG repeats that interact with Crm1 [28]. The 5 F-to-S mutations targeting the FG motifs that interact with Crm1 are shown as red vertical bars in the SQSTM1-NUP214^FGmut protein. (**B**) Intracellular localization of SQSTM1-NUP214, SQSTM1-NUP214^FGmut and endogenous Crm1 in MEFs transduced with HA-tagged SQSTM1-NUP214 and SQSTM1-NUP214^FGmut. Cells were stained with an anti-HA antibody and anti-Crm1 antibody and analyzed by confocal microscopy. Nuclei are shown with DAPI stain. Size bars, 20 μm. (**C, D**) Quantification of the relative binding of Crm1 to SQSTM1-NUP214 or SQSTM1-NUP214^FGmut by coimmunoprecipitation assays. MEFs stably transduced with HA-tagged SQSTM1-NUP214 (SN) or SQSTM1-NUP214^FGmut (SN^FGmut) were used to immunoprecipitate the fusion proteins with an anti-HA antibody; MEFs infected with an empty vector (MSCVpuro, EV) were used as control. Blots were analyzed by immunoblotting concurrently with anti-HA and anti-Crm1 antibodies. A representative immunoblot is shown (**C**, left panel). The ratios of the Crm1 signal/HA signal quantified using a Li-Cor Odyssey system are shown below the immunoblot. Quantification of 8 co-immunoprecipitation assays performed on 4 biological replicates is shown in the right panel (mean ± SD). Significance was calculated using the paired t-test, **** p<0.0001. Immunoprecipitation was also performed using MEFs treated with LMB (10 nM, 2 hours). Representative results are shown (**C**, left panel) and quantification from individual experiments is shown in panel **D** (Crm1 binding to SQSTM1-NUP214 and to SQSTM1-NUP214^FGmut is shown in the left and right panels, respectively). On average, LMB exposure reduced Crm1 binding to SQSTM1-NUP214, and SQSTM1-NUP214^FGmut by 58% (p = 0.003), and 52% (p = 0.02), respectively. Significance was calculated using the paired t-test. * p<0.05, ** p<0.01.

DMEM supplemented with 10% calf serum. MEFs were transduced by co-culture with filtered supernatant from Plat-E cells and 2 μg/ml polybrene for 3 consecutive days. On the third day, cells were seeded in triplicate flasks and expanded in the presence of puromycin (2.5 μg/ml). When specified, MEFs were treated with leptomycin B (LMB) (Sigma-Aldrich), 10 nM, for 2 hours prior to cell collection for coIP or ChIP experiments.

## Transduction of fetal liver and bone marrow progenitors

To assay the effects of SQSTM1-NUP214 and SQSTM1-NUP214^FGmut on colony formation, murine fetal liver hematopoietic progenitors were transduced with either MSCVpuro or

MSCV-IRES-eGFP vectors. Fetal liver cells were recovered from E12.5–14.5 B6(Cg)-Tyr$^{C-2J}$/J (B6JC) fetuses. Fetal livers were harvested from embryos, and cells were resuspended in RPMI supplemented with recombinant interleukin (IL)-3 and IL-6 (10 ng/ml), stem cell factor (SCF –100 ng/ml) and 20% fetal bovine serum (1 ml per fetal liver) by aspiration through a 25G nee- dle and cultured overnight in a 24 well plate. Fetal liver cells were transduced by spinoculation for 3 consecutive days with viral supernatant collected from transfected Plat-E cells as previ- ously described [24]. Transduced cells were seeded in methylcellulose media (HSC-CFU media, Miltenyi Biotec) enriched with IL-3, IL-6, granulocyte macrophage-colony stimulating factor (all at 10 ng/ml) and SCF (50 ng/ml) at a density of 100,000 cells/ml in a 6 well plate. Six days later, dense colonies consisting of >100 cells were counted. Cells harvested from the pooled colonies were serially replated to generate secondary and tertiary colonies.

Bone marrow progenitors were obtained from mice treated with 5-fluorouracil (150 mg/kg, tail vein injection) 4 days prior to harvest and further enriched in early progenitors by deple- tion of differentiated myeloid and B cells as previously described [24]. They were transduced using the same method described for the fetal liver cells. To perform qRT-PCR analysis of *Hoxa* gene expression, fetal liver or bone marrow hematopoietic cells transduced with MSCVpuro constructs were selected in RPMI media with 20% FBS, IL-3 (10 ng/ml), IL-6 (10 ng/ml), SCF (50 ng/ml), and puromycin (2.5 µg/ml) for 3 to 5 days prior to RNA recovery.

## Transplantation of syngeneic mice

All mouse studies were approved by the Duke University Institutional Animal Care and Use Committee. Recipient B6(Cg)-Tyr$^{C-2J}$/J (B6JC) mice were lethally irradiated (10 Gy in two split doses, 3 h apart) 24 h prior to transplantation. Mice were injected (via tail vein) with 650,000 transduced fetal liver cells along with 250,000 freshly harvested bone marrow cells to ensure hematopoietic recovery from irradiation. Mice were bled every two weeks by man- dibular vein puncture and blood was collected in EDTA tubes. Following red cell lysis using an ammonium chloride solution (Stemcell Technologies), blood samples were analyzed on a C6 flow cytometer (BD Biosciences Accuri™) to determine leukocyte concentration and per- centage of GFP expressing cells. Mice were monitored and euthanized by $CO_2$ inhalation when signs of disease were observed and all efforts were made to minimize animal suffering. Spleens were weighed and bone marrow was harvested by flushing tibias and femurs. Bone marrow cells were analyzed by flow cytometry to measure expression of GFP and surface markers Mac-1, Gr-1, c-Kit, and B220. RNA and protein were recovered from the bone mar- row, and residual cells were frozen in DMSO. To assay transplantability, sublethally irradi- ated (5 Gy) secondary recipients were injected with 5x $10^5$ to $10^6$ bone marrow cells from a primary diseased mouse.

## Immunofluorescence

MEFs infected with MSCV-IRES-eGFP vectors expressing HA tagged SQSTM1-NUP214 or SQSTM1-NUP214$^{FGmut}$ were grown on coverslips for 24 hours prior to processing. Cells were fixed with 4% paraformaldehyde then permeabilized with 0.1% Triton-X-100 in PBS. Cells were blocked for 30 minutes with 1%BSA, 22.52 mg/ml glycine in PBS-0.1% Tween-20 (PBS-T). Cells were stained with 1:100 primary antibody (anti-HA, Cell Signaling; anti-Crm1, Invitrogen) with 1% BSA in PBS-T for 18 hours. Anti-rabbit and anti-mouse secondary anti- bodies (ThermoFisher/Invitrogen) conjugated to Alexafluor 555 (to detect Crm1 in Fig 1) or to Cy5 (to detect HA-tagged fusion proteins in Fig 1) were used at 1/1000 dilution with 1% BSA in PBS-T, and incubated for 1 hour. Cells were counterstained with NucBlue (Thermo- Fisher/Invitrogen) for DAPI staining for 5 minutes. Three washes with PBS were performed

between each step. After adding mounting media (Vectashield, Vector Laboratories) coverslips were attached to glass slides with clear nail polish. All images were acquired on an Olympus Fluoview Fv1000 laser confocal with a 60X oil objective. All acquisition parameters including laser intensity and high voltage were maintained constant between samples. Brightness and contrast were adjusted with FIJI image analysis software. Quantification of the fraction of fluorescently labelled protein in the nuclear compartment was performed with the FIJI Intensity Ratio Nuclei Cytoplasm Tool plugin (NIH, USA).

## Co-immunoprecipitation and immunoblot assays

MEFs expressing SQSTM1-NUP214 or SQSTM1-NUP214$^{FGmut}$ and expanded in the presence of puromycin (2.5 μg/ml) were collected to prepare lysates from 15 to 20 million cells. Cells were trypsinized, rinsed in PBS and spun, the pellets were resuspended in cold Gough buffer (10 mM Tris-HCl pH 8, 150 mM NaCl, 1.5 mM MgCl2, 0.65% NP-40; 400 μl per 20 million cells) and incubated on ice for 12 minutes to lyse the plasma membranes. After centrifugation, the nuclear pellets were resuspended in lysis buffer C (20 mM Hepes pH 7.9, 25% glycerol, 400 mM NaCl, 1.5 mM MgCl2, 0.2 mM EDTA; 120 μl per 20 million cells) for 2 hours with agitation every 15 minutes as described [30]. Gough buffer and lysis buffer C were supplemented with protease inhibitor cocktail (Sigma P8340). Lysates were spun for 10 minutes and supernatants transferred to a new tube. Supernatants were diluted with 135 mM NaCl buffer (20 mM Tris pH 8, 1% glycerol, 135 mM NaCl, 0.5% NP-40) to bring the volume to 1 ml per sample. An aliquot (45 μl) was set aside for input before adding 2 μl of rabbit anti-HA-Tag antibody (Cell Signaling Technology C29F4) and incubation overnight at 4˚C with rotation. Protein G Dynabeads (Invitrogen) were added before incubation 2–4 hours at 4˚C with rotation. Beads were washed twice with 150 mM NaCl buffer followed by two washes with 175 mM NaCl buffer. Proteins were eluted by boiling pellets resuspended in 25 μl of Laemmli buffer for 20 min. Lysates were loaded on 7.5% acrylamide gels for SDS-PAGE and immunoblotting. Membranes were incubated simultaneously with rabbit anti-Crm1 (Bethyl A300-469A) and anti-HA (Cell Signaling Technology C29F4) antibodies. Results were visualized and quantified using an Odyssey imager (LI-COR).

Detection of HA-tagged fusion proteins in transduced cells and bone marrow from leukemic mice was performed with the anti-HA (Cell Signaling Technology C29F4) antibody. The mouse monoclonal anti-α-tubulin (Sigma T-9026) was used for loading control. Secondary anti-mouse and anti-rabbit horseradish peroxidase conjugated antibody (Invitrogen) were used to visualize the proteins by chemiluminescence.

## Immunophenotyping of colony forming cells

Fetal liver cells transduced with MSCVpuro vector, SQSTM1-NUP214 or SQSTM1-NUP214$^{FGmut}$ were grown in methylcellulose media with IL-3, IL-6, granulocyte macrophage-colony stimulating factor (all at 10 ng/ml) and SCF (50 ng/ml) and puromycin (2.5 μg/ml) to generate primary colonies. After 7 days the cells in methylcellulose media were rinsed in PBS and stained with the antibodies against Mac-1 (CD11b, PerCP/Cy5.5-conjugated (Biolegend)) and Gr-1 (Ly-6G/Ly-6C, APC-conjugated (Biolegend)). Cells were analyzed on an Accuri Flow cytometer (BD Biosciences) with CFlow Plus software (Accuri).

## Quantitative reverse transcriptase PCR (qRT-PCR)

RNA was recovered using RNeasy Mini kit (Qiagen) from MEFs or cultured hematopoietic progenitors selected in puromycin (2.5 μg/ml), or from bone marrow blasts harvested from leukemic mice. RNA was reverse transcribed using the iScript kit (Bio-Rad). Quantitative real

time PCR amplification was performed using the iQ Sybr Mix (Bio-Rad) with the CFX Connect System (Bio-Rad). Each cDNA was tested in triplicate. Gene levels were normalized to the *Gapdh* reference gene. *Hoxa9/Meis1* leukemias used as control to compare gene expression in bone marrow cells were previously described [24]. Primers (Table 1) mapping to the untranslated regions of *Hoxa9* and *Meis1* were used to avoid amplifying the transduced proviral sequences in *Hoxa9/Meis1* leukemia cells.

## Chromatin immunoprecipitation (ChIP) assays

Cross-linking was achieved by incubating MEFs (1x $10^7$ cells) in 1% formaldehyde at room temperature for 5 min, and the reaction was quenched with glycine (0.125 M). Fixed cells were rinsed in PBS and pellets were stored at -80 C. Cells were lysed in 1 ml of cold lysis buffer A (50 mM Hepes pH 7.5, 140 mM NaCl, 1 mM EDTA, 10% glycerol, 0.5% NUP-40, 0.25% Triton-X, 1x protease inhibitor cocktail (PI; Sigma P8340). After 10 min on ice, cells were pelleted and resuspended in 1 ml of cold buffer B (10 mM Tris-HCl pH 8, 200 mM NaCl, 1 mM EDTA, 0.5 mM EGTA, 1x PI) and incubated on ice for another 10 min. Cells were pelleted and resuspended in cold buffer C (10 mM Tris-HCl pH 8, 100 mM NaCl, 1 mM EDTA, 0.5 mM EGTA, 0.5% N-lauroylsarcosine, 1x PI) and sonicated to an average fragment size of 500 bp.

Immunoprecipitation was performed with rabbit anti-HA antibody (C29F4 Cell Signaling Technology) at 4˚C overnight. Protein G Dynabeads (ThermoFisher) were added and rocked at 4˚C for an additional 4 hours. Magnetic beads were washed once in 3 consecutive cold buffers: TSE 150 (10 mM Tris-HCl pH 8, 150 mM NaCl, 2 mM EDTA, 1% Triton-X, 0.1% SDS), TSE 500 (10 mM Tris-HCl pH 8, 500 mM NaCl, 2 mM EDTA, 1% Triton-X, 0.1% SDS) and wash buffer (10 mM Tris-HCl pH 8, 1 mM EDTA, 0.25 M LiCl, 0.5% NP-40), and twice in cold TE (20 mM Tris-HCl pH 8, 2 mM EDTA), before being eluted in elution buffer (1% SDS, 0.1 NaHCO$_3$) at 65˚C. Following RNAse A and proteinase K treatment, input and ChIP DNA were purified with a PCR purification kit (Qiagen) and amplified with specific primers (Table 2) using qRT-PCR. Amplification values were normalized to input.

## Statistical analyses

Comparison of binding of Crm1 to SQSTM1-NUP214 and SQSTM1-NUP214$^{FGmut}$, or to either fusion proteins in the absence or presence of LMB, as determined by co-immunoprecipitation, was done using the paired t-test (Graph Pad Prism8).

Gene expression measured by qRT-PCR was assumed to follow normal distribution and the statistical choices were based on previous reports [31]. Cells transduced and selected with puromycin in individual culture vials were considered as one biologic replicate (n = 1). To account for variability between experiments (randomized block experiments), a repeated-

**Table 1. Primers used for qRT-PCR.**

| Gene | Forward primer | Reverse primer |
|------|---------------|----------------|
| *Gapdh* | CCTGGAGAAACCTGCCAAGTATG | AGAGTGGGAGTTGCTGTTGAAGTC |
| *Hoxa7* | TATGTGAACGCGCTTTTTAGCA | GAAGTCGGCTCGGCATTTTG |
| *Hoxa9* | CCCCGACTTCAGTCCTTGC | GATGCACGTAGGGGTGGTG |
| *Hoxa10* | GGAAGCATGGACATTCAGGT | CCAGGCAAGCAAGACCTTAG |
| *Meis1* | TCAGCAAATCTAACTGACCAGC | AGCTACACTGTTGTCCAAGCC |
| *Hoxa9UTR* | GCGCTATTGGCTGTATGTGC | ATCTGCCACACGAAGAGCAA |
| *Meis1UTR* | TGGTAGTTGGGTCTGAGGGG | GCGTCGCTGAGCATAAAAGC |

**Table 2. Primers used for ChIP-PCR.**

| Gene | Forward primer | Reverse primer |
|------|----------------|----------------|
| Brg1 | CGCTTTTGCCGTACTTCTTC | TGAGACAAGGCAGCAGAGAA |
| Hoxa7 | GGTAGATGCGGAAACTGGCT | CGCCTCCTACGACCAAAACA |
| Hoxa9 | CCCCGACTTCAGTCCTTGC | GATGCACGTAGGGGTGGTG |
| Hoxa10 | TCTGGTGCTTCGTGTAAGGG | GCCTCGACTTAACCTTCCCC |
| Meis1 | GGAGGGAACAATGAGCCGAG | TGAGTTTCGTTCTCCAGCGG |

measures one-way analysis of variance (ANOVA) statistical test was used, followed by the Tukey multiple comparison test (Graph Pad Prism8).

A ratio paired t-test (Graph Pad Prism8) was used to determine the significance of the effect of LMB on the binding of SQSTM1-NUP214 and SQSTM1-NUP214$^{FGmut}$ to *Hoxa* and *Meis1* genes, as measured by ChIP.

## Results

### Targeting Crm1-interacting FG motifs impairs binding of SQSTM1-NUP214 to Crm1

We used site directed mutagenesis to exchange phenylalanines to serines in the 5 FG motifs previously shown to interact with CRM1 (Fig 1A). To compare the properties of SQSTM1-NUP214 and SQSTM1-NUP214$^{FGmut}$, we first retrovirally expressed the HA-tagged fusion proteins in murine embryonic fibroblasts (MEFs). We studied the intracellular distribution of the two proteins by confocal analysis of cells stained with an anti-HA antibody. As we previously reported in HeLa cells [10], SQSTM1-NUP214 displays a predominantly cytoplasmic punctate localization accompanied by fewer nuclear speckles. The cellular distribution of the SQSTM1-NUP214$^{FGmut}$ protein shows a similar pattern (Fig 1B). On average 15% of fluorescently-labelled SQSTM1-NUP214 was present in the nuclear compartment compared to 20% of SQSTM1-NUP214$^{FGmut}$ (Fig 1B, S1 Data). We did not notice any overt perturbation of the intracellular localization of endogenous Crm1 in cells expressing either SQSTM1-NUP214 or SQSTM1-NUP214$^{FGmut}$; Crm1 displays mostly diffuse nuclear staining with brighter signal at the nuclear rim (Fig 1B). We did not detect co-localization of Crm1 with either SQSTM1-NUP214 or SQSTM1-NUP214$^{FGmut}$ (S1 and S2 Videos).

To study the interaction of SQSTM1-NUP214 and SQSTM1-NUP214$^{FGmut}$ with Crm1, we performed co-immunoprecipitation assays in stably transduced MEFs. Lysates of puromycin selected cells were immunoprecipitated with an anti-HA antibody, and cells transduced with the empty MSCVpuro vector were used as a negative control. The relative binding of SQSTM1-NUP214 and SQSTM1-NUP214$^{FGmut}$ to the endogenous Crm1 protein was determined by normalizing the intensity of the Crm1 signal to that of HA-labelled fusion proteins measured by immunoblot analyses (Fig 1D). These studies demonstrate that the ability of SQSTM1-NUP214$^{FGmut}$ to interact with Crm1 is reduced by more than 50% compared with SQSTM1-NUP214.

Upon binding to NES-bearing cargo proteins, Crm1 undergoes conformational changes that enhance its interaction with NUP214 [28, 32]. As a result, blocking the binding of Crm1 to its endogenous NES substrates might be expected to reduce its affinity for SQSTM1-NUP214. To test for this possibility, we treated MEFs with leptomycin B (LMB), a selective and effective small molecule that binds to the NES binding groove of Crm1 and sterically blocks its interaction with NES-bearing proteins [33]. We found that LMB exposure (10

nM for 2 hours) reduces the ability of both SQSTM1-NUP214 and SQSTM1-NUP214$^{FGmut}$ to co-immunoprecipitate with Crm1 by more than 50% (Fig 1D and 1E). This indicates that genetically targeting the NUP214 FG repeats known to interact with Crm1 and pharmacologically blocking the Crm1/NES interaction have additive effects that reduce the interaction of SQSTM1-NUP214 with Crm1.

## The Crm1-interacting FG motifs of NUP214 are required for SQSTM1-NUP214 leukemogenesis

To determine the contribution of Crm1 to the transforming properties of SQSTM1-NUP214, we transduced murine hematopoietic progenitors with SQSTM1-NUP214 or SQSTM1-NUP214$^{FGmut}$. We first compared the effect of the fusion proteins on the self-renewal of hematopoietic colony forming cells. Hematopoietic precursors from mouse fetal liver were transduced and seeded in methylcellulose in the presence of myeloid growth factors to generate colonies that were serially replated (Fig 2A). Compared with cells transduced with an empty vector that exhausted all their clonogenic potential after the first passage, SQSTM1-NUP214 cells displayed robust colony forming ability upon second and third seedings (generating a mean of 123 and 57 colonies, respectively). In contrast, SQSTM1-NUP214$^{FGmut}$ cells formed minimal numbers of secondary and tertiary colonies (mean 8 and 2 colonies, respectively). This difference in transforming potential of SQSTM1-NUP214 and SQSTM1-NUP214$^{FGmut}$ was not attributable to differences in expression of the fusion proteins, as both displayed similar intensities upon immunoblot analysis of the transduced progenitors (Fig 2A, left panel). To determine the effect of the fusion protein on myeloid maturation, transduced cells from primary colonies were stained with antibodies against the markers of myeloid maturation Mac-1 and Gr-1. Compared to cells transduced with the empty MSCV-puro vector, a higher percentage of *SQSTM1-NUP214*-transduced cells were engaged in the myeloid lineage (Fig 2B) but the expression levels of both Mac-1 and Gr-1 among these myeloid cells were considerably reduced (Fig 2C), indicating that SQSTM1-NUP214 impairs myeloid maturation. With respect to Gr-1 intensity, the effect of SQSTM1-NUP214$^{FGmut}$ on myeloid maturation was less pronounced (Fig 2B and 2C).

To determine the effect of mutating the Crm1-interacting FG repeats on leukemogenicity, we transplanted irradiated syngeneic mice with hematopoietic progenitors transduced with an MSCV-IRES-eGFP vector encoding *SQSTM1-NUP214* (n = 6 mice) or *SQSTM1-NUP214$^{FGmut}$* (n = 7 mice). The reconstituted mice were bled every two weeks from day 21 onward following transplantation to monitor the percentage of circulating GFP-expressing leukocytes by flow cytometry. Both cohorts initially displayed a high percentage of transduced leukocytes (Fig 2D), indicating that engraftment was comparable and effective in both groups of mice. However, while the percentage of *SQSTM1-NUP214*-transduced leukocytes in the peripheral blood remained on average close to 80%, the percentage of *SQSTM1-NUP214$^{FGmut}$*-transduced leukocytes dropped rapidly to an average of 10% by 76 days post transplantation (Fig 2D). It is unclear if this reduction can be attributed to a detrimental effect of SQSTM1-NUP214$^{FGmut}$ on hematopoietic cells survival, transduction of hematopoietic precursors with short term engraftment potential, and/or decreasing expression of the proviral sequences over time. All of the *SQSTM1-NUP214* mice eventually developed acute myeloid leukemias between 150- and 350-days post transplantation, while none of the *SQSTM1-NUP214$^{FGmut}$* mice displayed signs of leukemia such as enlarged spleen or presence of GFP-expressing cells in the peripheral blood or bone marrow at the time of death (Fig 2E).

The *SQSTM1-NUP214* leukemias were characterized by splenomegaly (mean spleen weight ± SD: 0.30 ± 0.13 g, S1 Data), variable leukocytosis, and invasion of the bone marrow with *SQSTM1-NUP214* expressing blasts, as evidenced by GFP-positive cells representing at

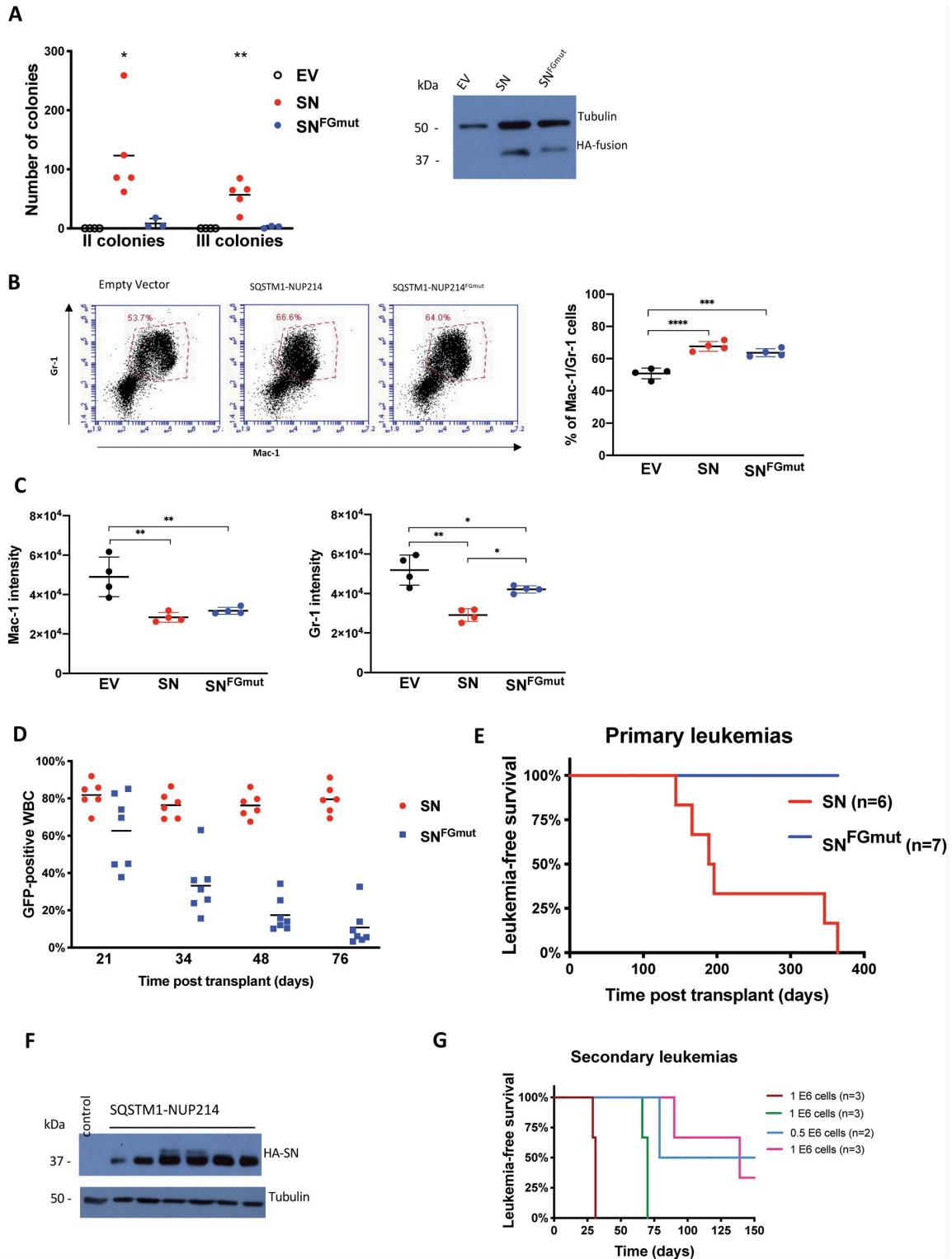

**Fig 2. Crm1-interacting FG repeats are necessary for the induction of leukemias by SQSTM1-NUP214.** (**A**) Colony forming assay of murine fetal liver hematopoietic progenitors transduced by empty vector (EV), *SQSTM1-NUP214* (*SN*) or *SQSTM1-NUP214*<sup>*FGmut*</sup> (*SN*<sup>*FGmut*</sup>). Each data point is the average number of secondary or tertiary passage colonies scored in at least 2 wells, resulting from the seeding of 10 000 cells. The means are shown from experiments repeated 3 to 5 times. The numbers of colonies were compared by one-way ANOVA with Tukey post hoc test. Significant differences are shown in comparison to cells transduced with the empty vector. *

p<0.05, ** p<0.01. Expression of the HA-tagged fusion proteins in fetal liver cells transduced with the indicated vectors is shown in the right panel. Protein lysates were prepared from unselected freshly transduced cells collected at the time of seeding to generate primary colonies. Tubulin was used as loading control. (**B**) Percentage of cells co-expressing the myeloid markers Mac-1 and Gr-1 in primary colonies. Cells transduced with the MSCV-puro vector (Empty vector), *SQSTM1-NUP214* or *SQSTM1-NUP214*^FGmut^ were grown in methylcellulose with puromycin for 7 days and harvested for co-staining with antibodies against Mac-1 and Gr-1 and analyzed by flow cytometry. Representative plots with the analysis gate used to define the population of double positive Mac-1/Gr-1 cells are shown. The graph shows the mean percentage of Mac-1$^+$/Gr-1$^+$ cells ±SD (4 replicates). (**C**) The mean fluorescence intensity of Mac-1 (left) and Gr-1 (right) expression in cells within the double positive population defined in (B) are shown. The graphs show the mean ±SD (4 replicates), one-way ANOVA was used to determine significance, * p<0.05, ** p<0.01, *** p<0.001, **** p<0.0001 (for B and C). (**D**) Percentage of GFP-positive white blood cells (WBC) in the peripheral blood harvested at the indicated time points after transplantation of mice with *SN* or *SN*^FGmut^-transduced fetal liver progenitors. The transduction efficiency of the fetal liver cells inoculated to the mice were comparable with 38% of *SQSTM1-NUP214*, and 43% of *SQSTM1-NUP214*^FGmut^-transduced cells expressing GFP. (**E**) Kaplan-Meier leukemia free survival curves of mice transplanted with *SN* or *SN*^FGmut^-transduced fetal liver progenitors. One of the *SN*^FGmut^ mice was euthanized at day 255 because of excessive weight loss and displayed an abdominal mass attached to the liver upon necropsy. All other *SN*^FGmut^ mice died of unknown cause, or were euthanized, between 364- and 466-days post transplantation. (**F**) Immunoblot analysis of SQSTM1-NUP214 protein expression in bone marrow cells from 6 *SQSTM1-NUP214* leukemic mice. The bone marrow of a mouse with a leukemia induced by *Hoxa9/Meis1* was used as a negative control. Tubulin was used as loading control. (**G**) Kaplan-Meier leukemia free survival curves of mice transplanted with 0.5 or 1 million bone marrow cells from 4 different primary *SQSTM1-NUP214* leukemic mice (n = 2 or 3 for each cohort). All mice were euthanized 150 days after transplantation.

least 95% of the bone marrow cells in terminally diseased mice. The majority of the bone marrow blasts expressed the myeloid marker Mac-1 (82 to 95%, S1 Data), while expression of the myeloid marker Gr-1 and the immature marker c-Kit were more variable (18 to 97%, and 13 to 49%, respectively, S1 Data). None of the leukemias expressed the B lymphoid marker B220. The SQSTM1-NUP214 protein was readily detectable in the bone marrow of diseased mice (Fig 2F). The *SQSTM1-NUP214* leukemias could be transplanted by injecting the bone marrow from a diseased mouse to sub-lethally irradiated recipients, although the latency and penetrance of these secondary leukemias were variable (Fig 2G).

Given that SQSTM1-NUP214 was identified in a leukemia with upregulation of *HOXA* genes [27], we examined the effect of SQSTM1-NUP214 on transcript levels of *Hoxa* and *Meis1*, an essential *Hox* cofactor, in the bone marrow blasts of murine leukemias induced by *SQSTM1-NUP214*. We used myeloid leukemias induced by the coexpression of *Hoxa9* and *Meis1* (*Hoxa9/Meis1*) as controls, since these genes recapitulate the disease while bypassing the upstream alterations that induce their overexpression [24, 34]. Compared to levels in *Hoxa9/Meis1* leukemias, endogenous *Hoxa9* and *Hoxa10* were considerably elevated in *SQSTM1-NUP214* leukemias (Fig 3A), whereas the levels of *Hoxa3*, *Hoxa5*, *Hoxa7*, *Hoxa11* and *Meis1* were comparable in *Hoxa9/Meis1* and *SQSTM1-NUP214* leukemias. These results suggest that *SQSTM1-NUP214* leukemias may be caused by the deregulation of specific *Hox* genes, such as *Hoxa9* and *Hoxa10*, which are validated drivers of leukemias [35, 36].

## The Crm1-interacting FG motifs of NUP214 contribute to the activation of *Hoxa* and *Meis1* genes in *SQSTM1-NUP214*-transduced hematopoietic cells

To further examine the potential contribution of *Hoxa* and *Meis1* genes to leukemogenesis, we studied their expression levels in hematopoietic cells freshly transduced with *SQSTM1-NUP214* or *SQSTM1-NUP214*^FGmut^. Hematopoietic precursors from either fetal liver or adult bone marrow cells were transduced with MSCVpuro vectors expressing either construct, selected in puromycin, and collected for qRT-PCR analyses. The levels of *Hoxa* and *Meis1* genes displayed a 2 to 4-fold increase in *SQSTM1-NUP214*-transduced cells compared with empty vector control cells (Fig 3B). While SQSTM1-NUP214^FGmut^ increased the expression of these genes, its ability to upregulate *Hoxa10* or *Meis1* in bone marrow and fetal liver cells was reduced compared to that of SQSTM1-NUP214 (Fig 3B). Overall, these results demonstrate that the ability of SQSTM1-NUP214 to transform hematopoietic cells correlates with its ability to increase the

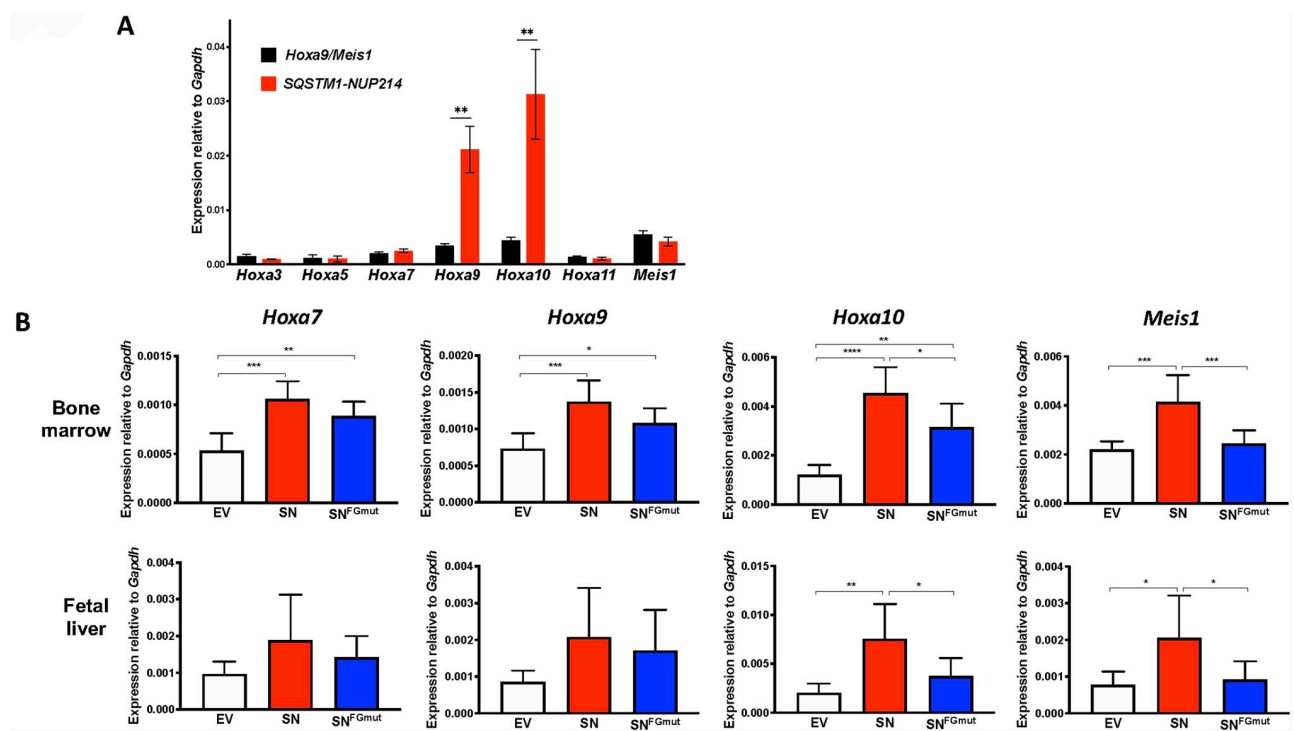

**Fig 3. SQSTM1-NUP214 induce the upregulation of *Hoxa* and *Meis1* genes in hematopoietic cells.** (**A**) Expression of *Hoxa* and *Meis1* genes determined by qRT-PCR in the bone marrow of leukemic *Hoxa9/Meis1* (black, n = 6) or *SQSTM1-NUP214* (red, n = 6) mice. Primers mapping to the untranslated regions of *Hoxa9* and *Meis1* were used to avoid amplifying the transduced proviral sequences in *Hoxa9/Meis1* leukemia cells. Shown are means ± SEM. Significance was calculated using the unpaired t test, ** p<0.01. (**B**) Murine hematopoietic cells from bone marrow (upper panels) or fetal liver (lower panels) were transduced with MSCVpuro (empty vector (EV), white), *SQSTM1-NUP214* (*SN*, red) or SQSTM1-NUP214$^{FGmut}$ (*SN$^{FGmut}$*, blue) and selected with puromycin for 3 to 5 days prior to RNA extraction. Transcript levels were determined by qRT-PCR and normalized to the reference gene *Gapdh*. Shown are mean ± SD from 6 (bone marrow) or 5 (fetal liver) independent experiments. Significance was calculated using the repeated-measures one-way ANOVA (Graphpad Prism). * p<0.05, ** p<0.01, *** p<0.001, **** p<0.0001.

expression of specific *Hoxa* and *Meis1* genes, indicating that these properties may be mediated by interaction with Crm1.

## Impaired Crm1 binding correlates with reduced SQSTM1-NUP214 occupancy of *Hoxa* and *Meis1* genes

To further investigate how SQSTM1-NUP214 upregulates the expression of *Hoxa* and *Meis1* genes, we examined MEFs transduced with either *SQSTM1-NUP214* or *SQSTM1-NUP214$^{FGmut}$* (Fig 4). Compared with cells transduced with the empty MSCVpuro vector, the expression of *Hoxa7*, *Hoxa9*, *Hoxa10* and *Meis1* was increased 30 to 60% in *SQSTM1-NUP214* MEFs (Fig 4A). The expression levels of these genes in *SQSTM1-NUP214$^{FGmut}$* MEFs were only marginally lower compared with *SQSTM1-NUP214* MEFs; a reduction in gene expression only reached statistical significance in the case of *Hoxa10*. To investigate whether SQSTM1-NUP214 directly activates the transcription of *Hoxa* and *Meis1* genes, we performed chromatin immunoprecipitation (ChIP). Whereas SQSTM1-NUP214 displayed minimal binding to a negative control gene (*Brg1*), we observed considerable enrichment of SQSTM1-NUP214 at *Hoxa7*, *Hoxa9*, *Hoxa10* and *Meis1* loci (Fig 4B). Compared with SQSTM1-NUP214, the binding of SQSTM1-NUP214$^{FGmut}$ showed a trend toward lower binding across all genes but the difference was not statistically significant. To explore whether further reducing the interaction of

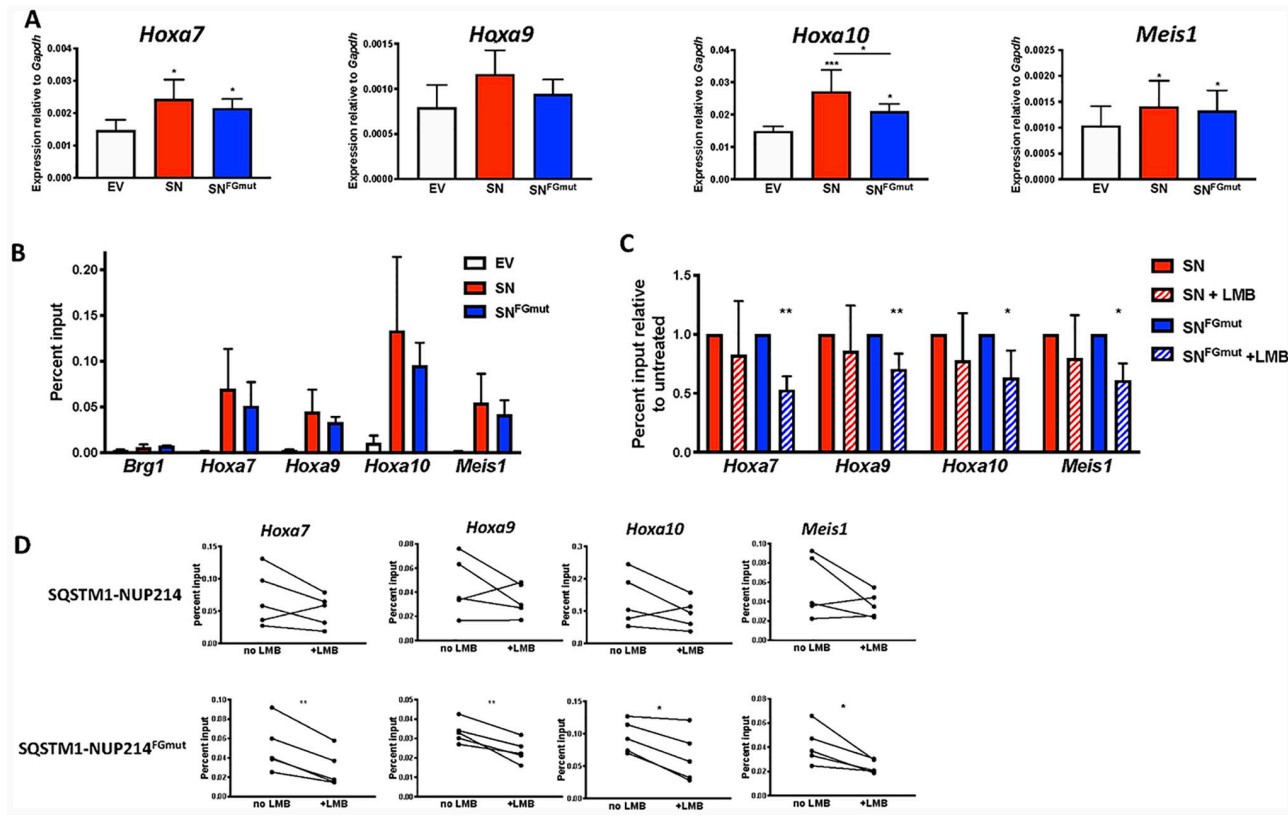

**Fig 4. SQSTM1-NUP214 directly activates *Hoxa* and *Meis1* genes in MEFs.** (**A**) Transcript levels in MEFs transduced with MSCVpuro (empty vector (EV), white), MSCVpuro encoding *SQSTM1-NUP214* (*SN*, red) or *SQSTM1-NUP214*^FGmut (*SN*^FGmut, blue). Cells were selected in puromycin 4 to 5 days prior to collection for RNA recovery. Expression was quantified by qRT-PCR and normalized to the reference gene *Gapdh*. Results shown are the mean ± SD compiled from 5 to 6 biological replicates (individually transduced MEFs). Significance was calculated using one-way repeated measures ANOVA followed by Tukey's multiple comparison test to account for experiment-to-experiment variability. (**B**) Chromatin Immunoprecipitation (ChIP) experiments performed with an anti-HA antibody to measure binding of HA tagged SQSTM1-NUP214 (red) or SQSTM1-NUP214^FGmut (blue) in transduced MEFs. Cells transduced with an empty vector (EV, white) serve as a negative control for the anti-HA antibody. The *Brg1* gene is used as a negative control. One-way ANOVA comparison did not show a significant difference in *Hoxa* and *Meis1* gene binding between SQSTM1-NUP214 and SQSTM1-NUP214^FGmut. Values shown are mean ± SD from 3 (Brg1 gene) or 5 biological replicates of transduced MEFs. (**C**) ChIP assays performed on MEFs grown in the absence or presence of LMB (10 nM, 2 hours). Results are shown normalized to untreated cells for clarity. Values shown are mean ± SD for 5 biological replicates. The significance of the effect of LMB was determined by a ratio paired t-test. * $p < 0.05$, ** $p < 0.01$. (**D**) Before and after line plots showing the effect of LMB on individual biological replicates (The averages are shown in (C)).

SQSTM1-NUP214 with Crm1 would reduce its ability to bind to *Hoxa* or *Meis1* genes, we repeated the ChIP on *SQSTM1-NUP214* and *SQSTM1-NUP214*^FGmut MEFs treated with LMB (10 nM for 2 hours). Whereas the reduction in binding of SQSTM1-NUP214 to chromatin caused by LMB was only on the order of 10–20% and did not reach statistical significance, LMB lowered the binding of SQSTM1-NUP214^FGmut to *Hoxa* and *Meis1* chromatin by 25 to 50% (Fig 4C and 4D). These results suggest that interfering with the ability of SQSTM1-NUP214 to interact with Crm1, by the additive effects of mutating NUP214 FG motifs and inhibiting binding of Crm1 to NES-containing cargo proteins (Fig 1D), compromises the ability of SQSTM1-NUP214 to occupy *Hoxa* and *Meis1* genes. These findings are congruent with a model by which Crm1 facilitates the recruitment of SQSTM1-NUP214 to its target genes.

## Discussion

Several lines of evidence point to a key role for Crm1 in the oncogenic properties of NUP98 and NUP214 fusion oncoproteins: NUP moieties of different fusion proteins always contain

FG motifs that mediate interaction with CRM1, and studies have shown that NUP fusion proteins cause mis-localization of CRM1 and impairment of CRM1-mediated nuclear export [11, 12]. The SQSTM1-NUP214 fusion protein offers a unique opportunity to test this assumption by introducing specific point mutations that target its ability to interact with CRM1. This was made possible by a crystal structure study that identified the specific NUP214 FG repeats that interact with CRM1 within a NUP214 fragment spanning amino acids 1916–2033 [28]. This NUP214$^{1916-2033}$ fragment contains 12 FG motifs, of which 9 interact with CRM1. The SQSTM1-NUP214 fusion protein contains the 122 carboxy terminal amino acids of NUP214 (amino acids 1969–2090) that include 14 FG motifs. The SQSTM1-NUP214$^{FGmut}$ was generated by targeting all 5 of the FG motifs positioned between amino acid 1969 and 2033 that had been previously shown to interact with CRM1 (Fig 1A). However, 6 of the 14 FG motifs present in SQSTM1-NUP214 are positioned beyond amino acid 2033 and their ability to interact with CRM1 or murine Crm1 is unknown. It is plausible that some of these untested FG motifs interact with Crm1 and could mediate the residual binding of SQSTM1-NUP214$^{FGmut}$ with Crm1 that we observed in MEFs (Fig 1C). We cannot formally rule out the possibility that the point mutations targeting the FG motifs critical for Crm1 binding do not also impair the interaction of SQSTM1-NUP214 with other proteins. In the context of NUP98, the FG repeat rich domain was also shown to interact with both CBP/p300 [16] and the NSL/MLL1 complex [23]. However, these previous studies were based on truncating large regions of NUP98, whereas our approach of precisely targeting the residues shown to interact with CRM1 would be expected to minimize off-target effects.

The molecular basis of how Crm1 contributes to the leukemogenicity of NUP98 and NUP214 fusion oncoproteins is not fully understood. While multiple studies have reported impairment of nuclear export by NUP fusion proteins, it is unclear how this might lead to the upregulation of *Hoxa* effector genes. We and others have shown that Crm1 can bind to the regulatory region of *Hoxa* and *Meis1* genes [20, 37, 38]. Similar to the role played by Crm1 in CALM-AF10 leukemogenesis [24], we propose that Crm1 enables the recruitment of NUP fusion proteins to the regulatory regions of *Hoxa* genes. We demonstrate here that impairing the ability of SQSTM1-NUP214 to bind Crm1, by mutating FG motifs and treating the cells with LMB, leads to decreased binding of the fusion protein to *Hoxa* and *Meis1* genes. While the 50% reduction in the ability of SQSTM1-NUP214$^{FGmut}$ to interact with Crm1 did not translate into an appreciable diminution in expression of *Hoxa* and *Meis1* genes in MEFs (Fig 4), this effect was much more pronounced in the context of hematopoietic progenitors (Fig 3), which are more relevant to leukemogenesis. The reason for the differences between MEFs and hematopoietic cells is unclear. We were unsuccessful in detecting binding of Crm1 and SQSTM1-NUP214 to *Hoxa* or *Meis1* loci by ChIP in hematopoietic cells, suggesting that the interaction of these proteins with chromatin may be more labile in hematopoietic cells compared to MEFs.

Studies of leukemogenesis by other NUP214 fusions have shown discrepant results. DEK-NUP214 can cause AMLs with overexpression of *HOXA* and *HOXB* in immunocompromised mice engrafted with transduced human CD34+ progenitors [39]. However, expression of SET-NUP214 impaired hematopoietic differentiation but did not cause leukemia in transgenic mice [40]. Our study is the first to examine the leukemogenic properties of SQSTM1-NUP214. The murine leukemias obtained by transplantation of hematopoietic precursors transduced with SQSTM1-NUP214 are myeloid and display upregulation of *Hoxa* genes. These features are consistent with the two clinical reports of SQSTM1-NUP214 leukemias that have been published to date, a patient with an immature phenotype T-ALL with upregulation of *HOXA* genes [27], and a patient with AML (in whom gene expression information was not determined) [26]. The correlation between the leukemogenic potential of SQSTM1-NUP214 and SQSTM1-NUP214$^{FGmut}$

and their ability to induce the expression of *Hoxa* and *Meis1* genes in hematopoietic cells (Fig 3) suggests that these genes are important effectors of leukemogenesis. Both the leukemia model and the *in vitro* transforming assay described herein will be useful to inform additional structure function analyses of SQSTM1-NUP214; in particular, it will be worth investigating which of the various domains within the SQSTM1 moiety are critical for leukemogenesis.

Our studies of SQSTM1-NUP214, along with the reports showing that Crm1 enables the binding of NUP98-HOXA9 and of SET-NUP214 to *HOXA* and *HOXB* target genes [20, 25], suggest that Crm1 could play a similar role with regards to other FG-NUP leukemogenic fusion proteins. Our results demonstrating that impairing the ability of SQSTM1-NUP214 to bind to Crm1 abrogates its leukemogenic potential point to the therapeutic potential of targeting the FG-NUP/Crm1 interaction. In this regard, Selective Inhibitors of Nuclear Export (SINEs, such as Selinexor) [41] that block the binding of Crm1 to NES could be beneficial, since the association of Crm1 with NES-containing proteins greatly enhances its binding to NUP214. Of note, a patient with AML associated with a DEK-NUP214 fusion displayed a remarkable response to treatment with Selinexor in a phase I clinical trial [42], underscoring the clinical relevance of the Crm1/NUP214 interaction. Given that inhibiting binding of Crm1 to NES cargo also impairs the interaction of Crm1 with NUP98 fusion proteins [20], SINEs could likewise be of therapeutic utility in the context of NUP98 fusion leukemias, which respond poorly to existing therapies.

In summary, we have demonstrated that the ability to interact with Crm1 correlates with the recruitment of SQSTM1-NUP214 to *Hoxa* and *Meis1* genes, the activation of their expression, and the induction of leukemias in mice. In conjunction with previous studies showing that Crm1 tethers CALM-AF10 [37], NUP98-HOXA9 [20], SET-NUP214 and mutated NPM1 [25] to their *HOX* target genes, this work points to a recurrent role for Crm1 in the deregulation of effector genes by leukemogenic oncoproteins.

## Supporting information

**S1 Data. Nuclear/Cytoplasmic distribution of SQSTM1-NUP214 and SQSTM1-NUP214FGmut and characteristics of SQSTM1-NUP214 leukemic mice.** Calculation of the nuclear/cytoplasmic ratio of the signals of SQSTM1-NUP214 (SN) and SQSTM1-NUP214FGmut (SNFG) based on confocal microscopy images analyzed with the FIJI Intensity Ratio Nuclei Cytoplasm tool plugin (first tab). Spleen weights and percentages of bone marrow cells expressing Mac-1, Gr-1 and c-Kit from diseased mice (second tab). (XLSX)

**S2 Data. Raw data used to generate the graphs shown in the figures.** Co-immunoprecipitation data for Fig 1 are shown in the first tab. Data for Fig 2 are shown in the second tab: number of secondary and tertiary colonies (Fig 2A), percentages of cells expressing both Mac-1 and Gr-1 in the colony assay (Fig 2B), and intensity of Mac-1 (Fig 2C left panel) or Gr-1 (Fig 2C right panel) in cells cultured in the colony assay. Data for Fig 3 are shown in the third tab: expression of *Hoxa* and *Meis1* genes in the bone marrow of individual leukemic mice (Fig 3A), and expression of *Hoxa* and *Meis1* genes in cultured bone marrow and fetal liver hematopoietic progenitors. Data for Fig 4 are shown in the fourth tab: expression of *Hoxa* and *Meis1* genes in MEFs (Fig 4A), ChIP results at *Brg1*, *Hoxa7*, *Hoxa9*, *Hoxa10* and *Meis1* loci (Fig 4B), ChIP results at *Hoxa7*, *Hoxa9*, *Hoxa10* and *Meis1* loci in cells treated with LMB shown relatively to untreated cells (Fig 4C), ChIP results at *Hoxa7*, *Hoxa9*, *Hoxa10* and *Meis1* loci without or with LMB treatment. (XLSX)

**S1 Raw images. Original uncropped and unaltered images underlying the blots shown in Figs 1D, 2A and 2F.**
(PDF)

**S1 Video. Video rendered with Imaris software of images obtained with a DeltaVision OMX-SIM microscope examining putative colocalization of Crm1 with SQSTM1-NUP214.** Crm1 is stained in green, SQSTM1-NUP214 (detected with an anti-HA antibody) is stained in magenta and DAPI is in blue, any co-localization of the fusion proteins with Crm1 would appear in white.
(MP4)

**S2 Video. Video rendered with Imaris software of images obtained with a DeltaVision OMX-SIM microscope examining putative colocalization of Crm1 with SQSTM1-NUP214$^{FGmut}$.** Crm1 is stained in green, SQSTM1-NUP214$^{FGmut}$ (detected with an anti-HA antibody) is stained in magenta and DAPI is in blue, any co-localization of the fusion proteins with Crm1 would appear in white.
(MP4)

## Acknowledgments

We are very grateful to Amanda Conway for her technical advice regarding co-immunoprecipitation and ChIP experiments. We also thank April Reedy and the Emory University Integrated Cellular Imagine Microscopy Core of the Emory+Children's Pediatric Research Center for assistance with confocal microscopy.

## Author Contributions

**Conceptualization:** Catherine P. Lavau, Daniel S. Wechsler.

**Data curation:** Waitman K. Aumann, Sei-Gyung K. Sze.

**Formal analysis:** Catherine P. Lavau, Waitman K. Aumann, Sei-Gyung K. Sze, Daniel S. Wechsler.

**Funding acquisition:** Daniel S. Wechsler.

**Investigation:** Catherine P. Lavau, Waitman K. Aumann, Sei-Gyung K. Sze, Veerain Gupta, Katelyn Ripple.

**Methodology:** Catherine P. Lavau, Ralph H. Kehlenbach, Daniel S. Wechsler.

**Project administration:** Daniel S. Wechsler.

**Resources:** Sarah A. Port, Ralph H. Kehlenbach, Daniel S. Wechsler.

**Supervision:** Catherine P. Lavau, Daniel S. Wechsler.

**Validation:** Catherine P. Lavau.

**Writing – original draft:** Catherine P. Lavau, Waitman K. Aumann, Daniel S. Wechsler.

**Writing – review & editing:** Catherine P. Lavau, Waitman K. Aumann, Sei-Gyung K. Sze, Ralph H. Kehlenbach, Daniel S. Wechsler.

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
