## [Decision Letter · Decision Letter 0]

24 Jan 2020

PONE-D-19-36059

The SQSTM1-NUP214 fusion protein interacts with Crm1 to activate Hoxa and Meis1 genes and drive leukemogenesis in mice

PLOS ONE

Dear Wechsler,

Thank you for submitting your manuscript to PLOS ONE. After careful consideration, we feel that it has merit but does not fully meet PLOS ONE’s publication criteria as it currently stands. Therefore, we invite you to submit a revised version of the manuscript that addresses the points raised during the review process.

Dear Dr.Wechsler,

your paper has been reviewed and the comments are now available for your consideration, one reviewer has expressed major concerns that require to be addressed before publication. He/she also asked to improve the quality of figures.

We would appreciate receiving your revised manuscript by Mar 09 2020 11:59PM. To enhance the reproducibility of your results, we recommend that if applicable you deposit your laboratory protocols in protocols.io, where a protocol can be assigned its own identifier (DOI) such that it can be cited independently in the future. For instructions see: http://journals.plos.org/plosone/s/submission-guidelines#loc-laboratory-protocols

We look forward to receiving your revised manuscript.

Kind regards,

Riccardo Alessandro

Academic Editor

PLOS ONE

Additional Editor Comments (if provided):

Dear Dr.Wechsler,

your paper has been reviewed and the comments are now available for your consideration, one reviewer has expressed major concerns that require to be addressed before publication. He/she also asked to improve the quality of figures.

Journal Requirements:

2. To comply with PLOS ONE submission requirements, in your Methods section, please provide additional information regarding the experiments involving animals and ensure you have included details on (1) methods of sacrifice, (2) methods of anesthesia and/or analgesia, and (3) efforts to alleviate suffering.

Reviewers' comments:

Reviewer's Responses to Questions

**Comments to the Author**

1. Is the manuscript technically sound, and do the data support the conclusions?

Reviewer #1: Partly

Reviewer #2: Partly

2. Has the statistical analysis been performed appropriately and rigorously? 

Reviewer #1: Yes

Reviewer #2: I Don't Know

3. Have the authors made all data underlying the findings in their manuscript fully available?

Reviewer #1: No

Reviewer #2: No

4. Is the manuscript presented in an intelligible fashion and written in standard English?

Reviewer #1: Yes

Reviewer #2: No

5. Review Comments to the Author

Reviewer #1: Lavau et al. study how an oncogenic NUP214 fusion protein (SQSTM1-Nup214 = SN) drives leukemogenesis.

The paper is nicely written and the figures are easy to interpret. The experiments are controlled well and performed rigorously. The data demonstrate that FG repeats are important to SN's transforming cells both in vitro and in vivo. Cancer effects correlate with the efficiency of SN binding to the Crm1 nuclear export receptor. Also, effects may be conferred through modulation of Hoxa and/or Meis1 gene expression.

My only major concerns relate to some degree of overstatement regarding the role of Crm1 in the pathway, and a few experimental issues that should hopefully be relatively easy to resolve.

Major Comments:

1. While there is a correlative link between SN cancer and SN-Crm1 binding (albeit addressed using different cell types), a direct link is not demonstrated. Thus, language should be more carefully crafted throughout including title, abstract (e.g., line 46), intro (e.g., line 129), results (e.g., lines 455, 472), and discussion (e.g., lines 523, 588). The authors do ultimately acknowledge that alternative factors could be responsible (lines 534-536) and this sentiment should be reinforced throughout.

2. Figure 1A and 1B. To test for interaction, it would make more sense to assess SQSTM1-NUP214 co-localization with Crm1 in these experiments. As shown not much is learned here.

3. No rationale is articulated to explain why co-IPs (Fig 1A) were performed in MEFs instead of the more relevant hematopoietic progenitors used for engraftment. Makes the suggested Crm1 link feel a bit tenuous, and it is later acknowledged that there are differential effects of wild-type and FGmut transgenes on gene expression in either cell type.

4. Figure 4C,D and lines 494-495- if the model is that SN activity is CRM1-dependent, FGmut should be less sensitive to LMB, not more. This discrepancy should be resolved (or at least discussed) and the model revised accordingly.

Minor Comments:

1. Line 41- “greatly” should be replaced with a more scientific term.

2. Line 105- sentence starting “In contrast…” seems unwarranted considering prior statement (line 98) that SET-NUP214 binds HOXA loci.

3. Line 116- “demonstrate”; better word “study”?

4. Lines 127-130- sentences here should better clarify nature of the results for the reader, e.g. binding assays done in MEFs in vitro; FGmut lost leukemogenicity in vivo. Plus, Crm1 binding not shown to be reason for leukemogencity, this was only established in MEFs.

5. Line 300- could more thoroughly explain to the reader why the crystal structure is relevant (i.e., that it directly showed which FG repeats bound Crm1).

6. Lines 332 and 341; data in these paragraphs is shown in both Figs 1D and 1E, should be referenced appropriately.

7. Lines 366-367. Comment seems inaccurate- differential effects of FGmut vs wild-type are really only apparent for GR-1 levels.

8. Lines 417-418, 422-425. These lines seem to refer to measurements (spleen size, GFP cells in BM, BM markers, B220) that are not included in the data.

9. Lines 451-452- statement is inaccurate; FGmut has a noticeable effect on all of these genes, is not “impaired” just has less of an effect relative to wild-type.

10. Line 489- word “modestly” is problematic- LMB had effects on both WT and FGmut SN, should just say what they were.

11. Line 555- is DNS allowed in this journal?

Reviewer #2: Lavau et al. investigate the importance of the binding of SQSTM1-NUP214 to Crm1 and the contribution of this physical interaction to leukemia pathogenesis. Authors show that interrupting this interaction can prevent mice from developing leukemia, likely through blocking the recruitment of SQSTM1-NUP214 to the promoter regions of Hoxa and Meis1 genes to regulate their expression.

Major concerns

1. Authors stated that the subcellular distribution of SQSTM1-NUP214 and SQSTM1-NUP214FGmut are same. For the images authors present, it is hard to judge whether their claims are correct or not. While the authors definitely need to improve image quality, the quantification for the images is needed. Moreover, the property of the two cell lines used for the two fusion proteins seems different in term of their nucleus size and morphology. Was this caused by the expression of the different forms of the fusion protein?

2. Authors did not provide an explanation or discussion on how SQSTM1-NUP214FGmut-transducted hematopoietic progenitors are eliminated from mice over time while SQSTM1-NUP214-transducted hematopoietic progenitors develop acute myeloid leukemia in irradiated syngeneic mouse model.

3. Authors need to provide evidence to document whether the proliferation and/or survival of hematopoietic progenitors are influenced after transduction with SQSTM1-NUP214 and SQSTM1-NUP214FGmut, respectively.

Minor points

1. The figures look blur. This may happen due to PDF file conversion or other causes. Please correct this.

2. Please use FG repeat units full form in abstract. Also, please give the full form of other abbreviations in manuscript such as SQSTM1, Hoxa, Meis1, etc.

6. PLOS authors have the option to publish the peer review history of their article (what does this mean?). If published, this will include your full peer review and any attached files.

Reviewer #1: No

Reviewer #2: No

---

## [Author Response · Author response to Decision Letter 0]

3 Apr 2020

April 1, 2020

Riccardo Alessandro

Academic Editor

PLOS ONE

Dear Dr. Alessandro,

We appreciate the opportunity to submit a revised version of our manuscript “The SQSTM1-NUP214 Fusion Protein Interacts with Crm1 to Activate Hoxa Genes and Drive Leukemogenesis in Mice” to PLOS ONE. We are grateful for the thoughtful insights of the reviewers and we have revised the manuscript according to their suggestions. 

Specific responses to reviewer comments (italicized) are indicated below in red font; specific changes to the manuscript/Figures are indicated in bold.

Journal Requirements:

*We have revised the manuscript to meet PLOS ONE’s style requirements, including ensuring that headings meet those requirements, and that files are named appropriately.

2. To comply with PLOS ONE submission requirements, in your Methods section, please provide additional information regarding the experiments involving animals and ensure you have included details on (1) methods of sacrifice, (2) methods of anesthesia and/or analgesia, and (3) efforts to alleviate suffering.

*We have provided additional details in the Methods section including details on methods of sacrifice and efforts to alleviate suffering (lines 179-180). We did not use any procedure requiring anesthesia/analgesia.

3. PLOS ONE now requires that authors provide the original uncropped and unadjusted images underlying all blot or gel results reported in a submission’s figures or Supporting Information files.

*We have included 5 pieces of supporting information that include:

• raw data characterizing nucleus/cytoplasmic ratios and leukemia characteristics (S1_raw-data);

• raw data supporting the graphs shown in the Figures (S2_raw_data);

• the original and uncropped images for the gels shown in Fig 1D, Fig 2A and Fig 2F (S3_raw_images);

• Video rendered with Imaris Software of images obtained with a DeltaVision OMX-SIM microscope examining putative colocalization of Crm1 with SQSTM1-NUP214. (S4_raw_video);

• Video rendered with Imaris Software of images obtained with a DeltaVision OMX-SIM microscope examining putative colocalization of Crm1 with SQSTM1-NUP214FGmut. (S5_raw_video).

*We have removed the phrase “data not shown” (Discussion, line 575) – these were negative data, and there was nothing to actually be shown

5. Please ensure that you refer to the Tables (Table of primers used for qRT-PCR:) and (Table of primers used for ChIP-PCR:) in your text as, if accepted, production will need this reference to link the reader to the Table.

*We have renamed the Tables (Table 1 and Table 2) referenced Tables 1 and 2 in the body of the manuscript.

Review Comments to the Author

Reviewer #1: 

Lavau et al. study how an oncogenic NUP214 fusion protein (SQSTM1-Nup214 = SN) drives leukemogenesis.

The paper is nicely written and the figures are easy to interpret. The experiments are controlled well and performed rigorously. The data demonstrate that FG repeats are important to SN's transforming cells both in vitro and in vivo. Cancer effects correlate with the efficiency of SN binding to the Crm1 nuclear export receptor. Also, effects may be conferred through modulation of Hoxa and/or Meis1 gene expression.

*We are grateful to the reviewer for these positive comments.

My only major concerns relate to some degree of overstatement regarding the role of Crm1 in the pathway, and a few experimental issues that should hopefully be relatively easy to resolve.

Major Comments:

1. While there is a correlative link between SN cancer and SN-Crm1 binding (albeit addressed using different cell types), a direct link is not demonstrated. Thus, language should be more carefully crafted throughout including title, abstract (e.g., line 46), intro (e.g., line 129), results (e.g., lines 455, 472), and discussion (e.g., lines 523, 588). The authors do ultimately acknowledge that alternative factors could be responsible (lines 534-536) and this sentiment should be reinforced throughout.

*We agree with the reviewer that some of our conclusions were overstated and have toned down the wording throughout the manuscript. Accordingly, we have made changes (marked in red font) to the Title, the Abstract (line 44-45), the Introduction (lines 128-131), the Results (line 463-465 and line 485-486), and the Discussion (lines 599-600).

2. Figure 1A and 1B. To test for interaction, it would make more sense to assess SQSTM1-NUP214 co-localization with Crm1 in these experiments. As shown not much is learned here.

*The immunofluorescence pictures were included to document that SQSTM1-NUP214 and SQSTM1-NUP214FGmut have similar distribution within the cell, to rule out that the loss of function of the mutant could be ascribed to gross mislocalization. To document this more precisely (as suggested by reviewer 2), we quantified the nuclear and cytoplasmic fractions of SQSTM1-NUP214 and SQSTM1-NUP214FGmut in stably transduced fibroblasts. We found that both proteins are predominantly localized in the cytoplasm with 15% and 20% of SQSTM1-NUP214 and SQSTM1-NUP214FGmut, respectively, residing in the nucleus. These results have been added to the revised version of the manuscript (lines 298-301 and supporting information S1). 

The immunofluorescence analysis of Crm1 localization within cells expressing either SQSTM1-NUP214 or SQSTM1-NUP214FGmut is shown in revised Fig 1B (we have condensed panels 1B and 1C from the previous version of the manuscript) to rule out the possibility that the fusion proteins could grossly alter the intracellular distribution of Crm1. As suggested by the reviewer, we used immunofluorescence to examine the co-localization of Crm1 with the fusion proteins. Using confocal microscopy we did not detect colocalization of Crm1 with either SQSTM1-NUP214 or SQSTM1-NUP214FGmut. This can be seen in short videos rendered with Imaris Software of images obtained with a DeltaVision OMX-SIM microscope that have been added as supporting information (S4_raw_video, S5_raw_video). These have been referenced in the text (lines 304-305). The discrepancy between the co-IP and the co-IF likely results from differences in sensitivity between the two techniques used; possibly because the proteins interact within structures that are too small/faint to be detected by immunofluorescence.

3. No rationale is articulated to explain why co-IPs (Fig 1A) were performed in MEFs instead of the more relevant hematopoietic progenitors used for engraftment. Makes the suggested Crm1 link feel a bit tenuous, and it is later acknowledged that there are differential effects of wild-type and FGmut transgenes on gene expression in either cell type.

*We agree with the reviewer that it would be more relevant to assay the binding between Crm1 and SQSTM1-NUP214 or SQSTM1-NUP214FGmut in hematopoietic cells. The main reason for using cultured fibroblasts to perform the coIP experiments was the ease of obtaining large numbers of cells (15 to 20 million cells per sample); it would have been much more challenging to obtain such numbers of hematopoietic progenitors purified from fetal liver or bone marrow. Furthermore, because of the inherent variability of the assay, the coIP was repeated numerous times (n=9). This was greatly facilitated by using immortalized MEFs; it would have been much more burdensome to each time have to isolate primary hematopoietic cells from mice, to transduce, select and expand them in the presence of costly growth factors (the primary hematopoietic progenitors transduced with either SQSTM1-NUP214 or SQSTM1-NUP214FGmu can only be grown for a few days). We also chose fibroblasts because we had experience performing co-immunoprecipitations of endogenous Crm1 in these cells. We made the assumption that the cellular context should not considerably impact the biochemical interaction between SQSTM1-NUP214 and Crm1. 

4. Figure 4C,D and lines 494-495- if the model is that SN activity is CRM1-dependent, FGmut should be less sensitive to LMB, not more. This discrepancy should be resolved (or at least discussed) and the model revised accordingly.

*We are not sure we understand the reviewer’s concern. According to our model, Crm1 participates in the recruitment of SQSTM1-NUP214 to Hoxa/Meis1 target genes. When the residues that mediate Crm1 interaction are mutated, the interaction of SQSTM1-NUP214 with Crm1 is not abolished but only weakened (reduced by 50%, according to our coIP results, Fig 1D, E). We show that exposure to LMB, which alters the conformation of Crm1 (indirectly by blocking binding to NES substrates) to one less favorable to NUP214 interaction, further reduces the interaction of NUP214FGmut with Crm1 (Fig 1D, E). This additive effect of genetic and pharmacologic interference with the SQSTM1-NUP214/Crm1 interaction correlates with reduced binding of SQSTM1-NUP214 to Hoxa/Meis1 genes. We do not know why the relative effect of LMB on SQSTM1-NUP214FGmut appears to be more pronounced than that on SQSTM1-NUP214 in the ChIP assay. 

Minor Comments:

1. Line 41- “greatly” should be replaced with a more scientific term.

*We have changed “greatly” to “considerably” (Line 40).

2. Line 105- sentence starting “In contrast…” seems unwarranted considering prior statement (line 98) that SET-NUP214 binds HOXA loci.

*We have changed the sentence to “In contrast to NUP98 fusions, the mechanisms underlying the recruitment of NUP214 fusion proteins to HOXA genes have yet to be elucidated.” (Line 103).

3. Line 116- “demonstrate”; better word “study”?

*We changed “demonstrate” to “study” (Line 114).

4. Lines 127-130- sentences here should better clarify nature of the results for the reader, e.g. binding assays done in MEFs in vitro; FGmut lost leukemogenicity in vivo. Plus, Crm1 binding not shown to be reason for leukemogenicity, this was only established in MEFs.

*We have added details to better describe the experiments conducted in this study (Lines 127-131).

5. Line 300- could more thoroughly explain to the reader why the crystal structure is relevant (i.e., that it directly showed which FG repeats bound Crm1).

*We added that the crystal study “identified the FG repeats that interact with Crm1” (Line 311).

6. Lines 332 and 341; data in these paragraphs is shown in both Figs 1D and 1E, should be referenced appropriately.

*We added a reference in the text to Fig 1D since the effect of LMB is shown on the WB (Line 341).

7. Lines 366-367. Comment seems inaccurate- differential effects of FGmut vs wild-type are really only apparent for GR-1 levels.

*We have specified that the difference between FGmut and wild type is seen in terms of Gr-1 levels. (Line 374).

8. Lines 417-418, 422-425. These lines seem to refer to measurements (spleen size, GFP cells in BM, BM markers, B220) that are not included in the data.

*The characteristics of the SQSTM1-NUP214 leukemias were only described in the text. We have added the spleen weights of the leukemic mice and the percentages of bone marrow cells expressing Mac-1, Gr-1 and c-Kit in the supporting information S1 (Lines 430-31 and 434-436). 

9. Lines 451-452- statement is inaccurate; FGmut has a noticeable effect on all of these genes, is not “impaired” just has less of an effect relative to wild-type.

*We have modified the sentence to “While SQSTM1-NUP214FGmut increased the expression of these genes, its ability to upregulate Hoxa10 or Meis1 in bone marrow and fetal liver cells was reduced compared to that of SQSTM1-NUP214” (Lines 463-465).

10. Line 489- word “modestly” is problematic- LMB had effects on both WT and FGmut SN, should just say what they were.

*We changed the sentence to “Whereas the reduction in binding of SQSTM1-NUP214 to chromatin caused by LMB was only on the order of 10-20% and did not reach statistical significance, LMB lowered the binding of SQSTM1-NUP214FGmut to Hoxa and Meis1 chromatin by 25 to 50% (Fig 4C and 4D).”(lines 502-504).

11. Line 555- is DNS allowed in this journal?

*We were unaware that PLoS did not allow references to “data not shown”, we apologize for the oversight. We have removed this reference.

Reviewer #2:

Lavau et al. investigate the importance of the binding of SQSTM1-NUP214 to Crm1 and the contribution of this physical interaction to leukemia pathogenesis. Authors show that interrupting this interaction can prevent mice from developing leukemia, likely through blocking the recruitment of SQSTM1-NUP214 to the promoter regions of Hoxa and Meis1 genes to regulate their expression.

Major concerns:

1. Authors stated that the subcellular distribution of SQSTM1-NUP214 and SQSTM1-NUP214FGmut are same. For the images authors present, it is hard to judge whether their claims are correct or not. While the authors definitely need to improve image quality, the quantification for the images is needed. Moreover, the property of the two cell lines used for the two fusion proteins seems different in term of their nucleus size and morphology. Was this caused by the expression of the different forms of the fusion protein?

*We acknowledge the reviewer’s concerns regarding the immunofluorescence images provided in the original manuscript. The apparent difference in size of cells expressing SQSTM1-NUP214 or the FG mutant was due to inconsistencies in the enlargement of the pictures shown; we apologize for this error. As suggested by the reviewer we have repeated the IF experiments using confocal microscopy and have quantified the nuclear/cytoplasmic distribution of the two proteins. The results included in the revised manuscript (lines 298-301 and supporting information S1) indicate that the proportion of SQSTM1-NUP214FGmut located in the nuclear compartment is slightly higher than that of SQSTM1-NUP214 (20 vs. 15%). 

2. Authors did not provide an explanation or discussion on how SQSTM1-NUP214FGmut-transducted hematopoietic progenitors are eliminated from mice over time while SQSTM1-NUP214-transducted hematopoietic progenitors develop acute myeloid leukemia in irradiated syngeneic mouse model.

*We do not know why the number of cells transduced with SQSTM1-NUP214FGmut detected in the peripheral blood of the transplanted mice diminishes over time. It is unclear if this is caused by: 1) a detrimental effect of SQSTM1-NUP214FGmut on the survival of the hematopoietic cells; 2) limited retroviral transduction to hematopoietic precursors with short term engraftment potential; and/or 3) decreasing expression of the proviral sequences encoding SQSTM1-NUP214FGmut over time. These possible explanations have been added to the text (lines 423-426). 

3. Authors need to provide evidence to document whether the proliferation and/or survival of hematopoietic progenitors are influenced after transduction with SQSTM1-NUP214 and SQSTM1-NUP214FGmut, respectively.

*To conduct the qRT-PCR analyses of Hoxa and Meis1 gene expression, we grew transduced fetal liver and bone marrow progenitors in suspension culture with cytokines for 3-5 days. In those conditions, we did not observe any growth advantage of the cells transduced with either SQSTM1-NUP214 or the FG mutant, compared to cells transduced with the empty vector. The proliferation of the cells was short lived with very few viable cells persisting after 1 to 2 weeks of culture. Instead we showed using the serial replating colony forming assay that SQSTM1-NUP214, unlike the FG mutant, causes the sustained proliferation of colony forming cells. This difference in proliferation potential of SQSTM1-NUP214-transduced hematopoietic precursors was corroborated by the development of leukemias in transplanted mice.

Minor points:

1. The figures look blur. This may happen due to PDF file conversion or other causes. Please correct this.

*We apologize for the blurry aspect of the figures. The files that were uploaded on the journal website were at high resolution. By clicking on the link in the upper right hand corner of the Figure, a high-resolution image of the Figure downloads appropriately and is clear.

2. Please use FG repeat units full form in abstract. Also, please give the full form of other abbreviations in manuscript such as SQSTM1, Hoxa, Meis1, etc.

*We apologize for this oversight; we have added the full names of the acronyms either in the abstract or the text of the manuscript.

 Again, we appreciate the opportunity to revise our manuscript for PLOS ONE. We hope that we have sufficiently addressed the reviewers’ comments and concerns. Please do not hesitate to contact us if you need any additional information.

Sincerely,

Catherine P. Lavau, DVM, PhD Daniel S. Wechsler, MD, PhD

Assistant Professor Thomas R. Giddens Professor

Duke University Aflac Cancer & Blood Disorders Center

 Children’s Healthcare of Atlanta

 Emory University

---

## [Editor Report · Decision Letter 1]

7 Apr 2020

The SQSTM1-NUP214 fusion protein interacts with Crm1, activates Hoxa and Meis1 genes, and drives leukemogenesis in mice

PONE-D-19-36059R1

Dear Dr.Wechsler,

We are pleased to inform you that your manuscript has been judged scientifically suitable for publication and will be formally accepted for publication once it complies with all outstanding technical requirements.

With kind regards,

Riccardo Alessandro

Academic Editor

PLOS ONE

Additional Editor Comments (optional):

Authors have addressed all the concerns raised by the reviewers and the manuscript is now ready for publication in PLOS ONE.
---

## [Editor Report · Acceptance letter]

14 Apr 2020

PONE-D-19-36059R1 

The SQSTM1-NUP214 fusion protein interacts with Crm1, activates *Hoxa* and *Meis1* genes, and drives leukemogenesis in mice 

Dear Dr. Wechsler:

I am pleased to inform you that your manuscript has been deemed suitable for publication in PLOS ONE. Congratulations! Your manuscript is now with our production department. 

With kind regards,

on behalf of

Prof Riccardo Alessandro 

Academic Editor

PLOS ONE